# The differences of muscle activation in forehand serve-receiving technique of male tennis players at different skills

Yuxin Zhang[1], Jiajie Tian[2], Zhouye Chen🄳[1]*

1 Sport School of Athletic Performance, Shanghai University of Sports, Shanghai, China, 2 China Table Tennis College, Shanghai University of Sport, Shanghai, China

* 1163405922@qq.com

## Abstract

The purpose of this study was to analyze differences in muscle activation of the right upper limb and part of the trunk and differences in stroke performance of forehand receiving in male tennis players of different levels of performance at different serve speeds. Thirty male tennis players (no difference in age, height, weight) were divided into senior (AG, n=15) and intermediate (IG, n=15) groups to perform 6 forehands receive tests at 2 serve speeds: low-speed serve (130-140km/h), high-speed serve (160-170km/h). Muscle activity from the right of biceps brachii (BB), triceps brachii (TB), brachioradialis (BC), deltoid (DT), trapezius (TP), pectoralis major (PM), obliquus externus abdominis (OEA) and latissimus dorsi (LD) were recorded using surface electromyography during the concentric phase of the lift and expressed as a percentage of each muscle's maximal activity, recorded during a maximal isometric contraction. Returned speed and placement were recorded using a high-speed camera. The results showed that the AG had significantly lower muscle activity ($p < 0.05$) in the backswing, impart, follow-through phases of the receive and more consistent dominant muscles in all phases, while the opposite was true for the IG. At both serve speeds, AG had significantly faster ball speeds ($p < 0.05$) and higher placement scores ($p < 0.05$) compared to IG. It is important to develop the athlete's receive action at lower muscle activation and higher joint kinetic energy, which may be an important way to improve receive performance in a short period.

## Introduction

The serve-return has become 1 of the 2 most important shots in the modern tennis game [1]. With speeds over 200km/h and multiple serve trajectories, the serve-return player must react as quickly as possible. The effectiveness of the serve is directly dependent on the receiver's return technique. On hard courts, where serves are faster and more difficult to return [2,3], however, since hard courts (about 40%) return

**Data availability statement:** This work was funded by Shanghai Key Lab of Human Performance (Shanghai University of Sport) (NO. 11DZ2261100 to ZYC). The authors have no conflict of interest to declare.

**Funding:** The author(s) received no specific funding for this work.

**Competing interests:** The authors have declared that no competing interests exist.

more than red clay (about 30%) [4], understanding its muscle activation properties can help to improve the performance of tennis players' serve-receive returns.

The forehand serve-return is a complete power chain movement, Force production in tennis and other sports typically involves the transfer of ground reaction forces through the lower extremities and trunk to the upper extremities [4], with the arm and wrist being the end links that accomplish the transfer of power [5,6]. The mechanics of the forehand receive are similar to the forehand groundstroke, but due to the faster speed of the serve, the serve-returners required the stroke to be completed in a shorter period and the return speed is affected by the speed of the serve. Therefore, the receiving movement has some differences from the forehand groundstroke movement.

There are differences in muscle activation characteristics of athletes under different stroke conditions, for example: grip, stroke path, ball spin, and stroke technique [7–10]. In tennis strokes, trunk and upper limb muscle activation follow a proximal to distal progression, and studies have shown that the order of muscle activation is not related to the player's age or skill level. Rogowski concluded that with increasing racket mass, trunk and upper limb muscle temporal activation and deactivation do not change [11,12], suggesting that the order of muscle activation in forehand stroke and racket weight are not the main factors affecting stroke effectiveness. In addition, a study analyzing each phase of the forehand stroke found that the activation levels of the major muscle groups of the shoulder and upper limb were lower during the backswing phase compared to the high level of activation during the stroke phase and the medium level of activation during the follow-through phase [13,14]. By analyzing the electromyographic activation characteristics of the receiving technique, it was possible to analyze the differences in the level of muscle activation during the different stroke phases.

In addition, proper receive movements are beneficial in preventing muscle injuries, and Aben et al. point out that most injuries in tennis players are caused by single-segment overuse, which leads to the loading of the leading muscle groups, and long-term training produces pathologic changes [15]. In tennis, the elbow works as a connector in the kinetic chain, receiving energy and transmitting it to distant segments [16]. At different serve speeds, the receiving player's arm swings at different speeds, generating different pressures, the ability of the elbow and wrist joints to transmit and adjust forces may change, and the arm muscles may take on specific activity characteristics [17]. Increasing the speed of movement also leads to premature activation of the active and antagonist muscles of the upper limb [18,19]. Some studies have shown that elbow injuries are in the top three of tennis players' injuries [20], and one of the most commonly accepted theories about the elbow in tennis players is based on the stiffness of the skeletal muscle units and their repetitive overuse in tennis strokes. To date, there is no consensus on the leading etiology of elbow tendinopathy, although it has been suggested that it is multifactorial [7] and not gender-specific [21].

The forehand receive technique can be divided into various types of ways according to the compulsion degree of the receiving player from small to large, and the smash serve is the technique with a high use rate for the player to break the opponent's offense state [22,23], the effectiveness of the service depends on how well

the serve-return is done, therefore, the forehand smash receive was selected as the test technique in this study, excluding volley, slice. Based on the statistics of first and second-serve speeds of elite athletes, we set the slow-serve to 130–140km/h, and the high-serve to 160–170km/h [24,25]. A study was conducted on the statistics of serve placement of high-level tennis players who issued ACEs in matches [26], this study found that serve placement closer to the sideline on the serving side was more conducive to creating an advantage on the next shot and executing the technical play, and it was found that the serve placement of high-level players was highly concentrated in T-type (near the centerline of the serve) or wide-type (near the singles sideline of the serve area). When serving placement closer to the sideline, the receiver is more adept at using a forehand receive to enhance the speed and return placement. The receivers were more likely to use forehand techniques to receive wide-type serves to improve the speed and landing of the return [27,28]. The International Tennis Federation (ITF) and the State General Administration of Sport of China have developed the Provisional Determination of Sport Skill Levels [29], the first serve placement is used as an indicator of the quality of the serve, and the highest score is achieved when the first placement is within 1m² of the effective area. Therefore, in this study, the serve placement was set as wide-type serves (Fig 1). Understanding the similarities and differences in the muscle activation characteristics of forehand smashes received in high-level tennis players at different serve speeds is beneficial for improving the receive performance of the tennis player.

Currently, there is no EMG data on the muscle activation of the forehand serve using the forehand smash serve technique in athletes of different skill levels. Therefore, exploring the muscle activation levels of tennis players of different skill levels using forehand smash serve receive at different serve speeds can help to recognize the right arm and trunk muscle activation characteristics of the receive maneuver, prevent sports injuries, and improve the muscular strategies of athletes forehand receive at different serve speeds.

## Methods

### Participants

We used Gpower (α = 0.05, Power = 0.8, Effect size = 1.06) to determine the 24 participants in the experiment [30]. Fifteen advanced group (AG: Chinese national level 1) and Fifteen intermediate group (IG: Chinese national level 2) male

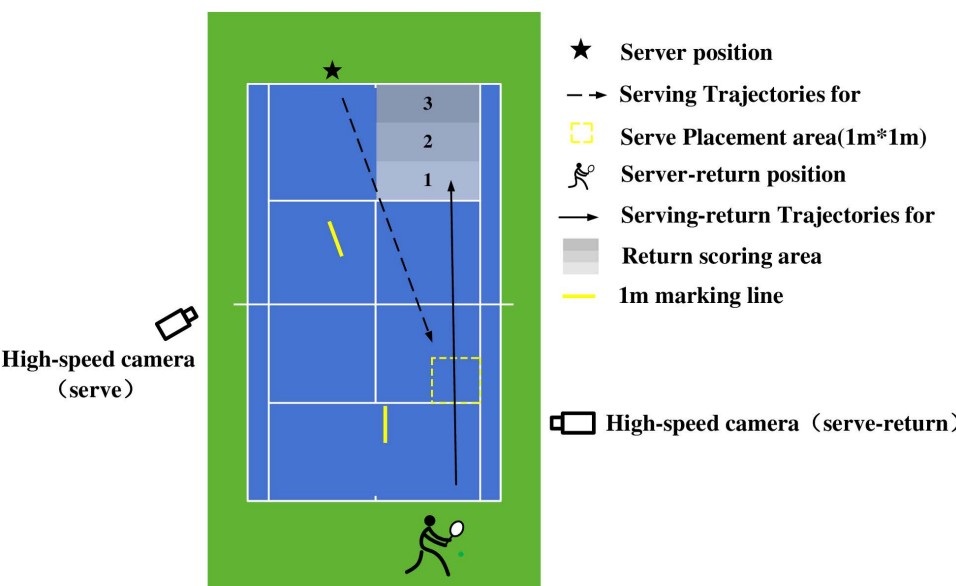

**Fig 1. Serve-return Motion Capture setup diagram.**

tennis players were selected to participate in the study. Height, weight, age, and training experience were compared between the two groups of players. There were no differences in all conditions except for training experience ($p > 0.05$, Table 1). All participants were right-handed and had no musculoskeletal injuries in the last six months. Before the start of the study, participants were informed about the design of the study, with a special emphasis on the possible risks and benefits, and all participants provided written informed consent before the enrolment. The study followed the guidelines of the Declaration of Helsinki and was approved by the Scientific Research Ethics Committee of Shanghai Sport University. The recruitment of the participants began on May 23, 2024, and ended on May 30, 2024 (document code: No.102772024RT008).

## Experiment protocol

All tests will be conducted at the tennis courts of Shanghai Sport University. The Delsys-16 EMG system (Delsys Inc., Natick, MA, USA) was employed to collect the electromyographic data of participants forehand receive movement. The frequency was set at 2000 Hz for sampling. The muscle activation signals of the biceps brachii (BB), triceps brachii (TB), brachioradialis (BC), deltoid (DT), trapezius (TP), pectoralis major (PM), obliquus externus abdominis (OEA) and latissimus dorsi (LD) were recorded in both upper extremities.

The current serve machine cannot achieve the speed, ball placement, and spin of the player's serve. Therefore, in this study, we chose the player with the highest level among the participants to serve. To ensure the accuracy of serving speed and placement. Before the formal experiment, we trained the server several times in advance and used high-speed cameras to test the speed of each ball to improve the accuracy of the serve. In the formal experiment, we regarded the data that the serve speed meets the requirements as valid data, otherwise excluded. Therefore, two serve speeds can be minimally guaranteed, and it is reasonable to compare the differences in the players' serve-receiving movements at the two serve speeds while ensuring that the serving conditions are consistent. The low-speed balls are kept between 130–140km/h and high-speed balls are kept between 160–170km/h. A wide-type placement field of 1m$^2$ in the right outer corner of the right serve was selected as the test area for this study (Fig 1). A 1m bright-colored marking line is set parallel to the direction of motion of the serve, and a high-speed camera is placed perpendicular to the 1m marking line to record the serve's motion at 1000Hz.

The subject stood to receive a serve in the right zone of the tennis court. Participants use a customary grip and a forehand receive technique that requires participants to stroke the ball in a straight line to the same side of the court with maximum effort. A 1m brightly colored marker line was set up parallel to the direction of movement of the return and a high-speed camera was set up perpendicular to the 1m marker line to record the return's movement at 1000Hz to capture the speed and placement of the returned ball (Fig 1).

The racket movement process was recorded using a Casio (High-speed camera, Japan, 1000fps), and racket head speed was calculated using the Kinovea program. Participants all used the same racket (grip 41/4 inches and mass 300g; Pure Drive; Babolat Play, France) with a reflective marker attached to the tip to record racket kinematics. The formal experiment was initiated after participants were given time to familiarize themselves with the process and the

**Table 1. Characteristic of the groups (mean±SD).**

| Variables | AG (n=15) | IG (n=15) | Cohen's d |
|---|---|---|---|
| Age (yr) | 23.07±2.40 | 22.88±2.53 | 0.07 |
| Height (cm) | 180.07±6.24 | 178.25±6.37 | 0.28 |
| Mass (kg) | 73.43±8.95 | 70.00±9.13 | 0.37 |
| Training experience (yr) | 12.29±3.65 | 5.94±1.35 | 2.30 |

Note: AG, Chinese national level 1; IG, Chinese national level 2.

environment, particularly the racket that was similar in mass and grip size to that typically used. Each athlete is tested with the new ODEA-passion (Zhejiang, China) ball.

## Procedure

Participants avoided any strenuous exercise for 24 hours before the study. Players performed a 10-minute slow running session on the field wearing athletic clothing and tennis shoes. Jogging was followed by approximately 5 minutes of static stretching activities and 10 minutes of forehand stroke practice to ensure that each participant could perform to their best potential during the experiment.

In preparation for the experiment, the skin of the participants' upper extremities was cleaned with alcohol before EMG electrodes were attached using sports foam and sticky tape. All of the sensor muscle placements were based on procedures outlined in Criswell [31].

For electromyographic data collection, the participant stood to receive a serve in the right zone of the tennis court. Fig 1 shows serve placement zones and return routes. Data were considered valid if both of the following conditions were met: 1) the serve ball speed was within the low/high-speed range and the placement was in bounds,2) the return direction was straight and the landing point was in bounds. Participants were tested with a low-speed serve followed by a high-speed serve, with three valid data collected at each serve speed and a 30-second rest period for each test, as individual fatigue could cause inaccuracies in data collection. We found that all participants used a closed stance for the received by observing the test video. The data collected only referred to the right of the trunk and arm. In this study, the maximum activation values of each muscle when the athletes did the test action were selected as maximal voluntary contractions (MVC) [32].

## Data analysis

In this study, the forehand serve-receiving can be divided into three phases: T1: Backswing phase: the phase was defined from when a player assumed the preparatory posture to the end of the backswing; T2: Impacting phase: the process was defined from the end of the backswing to ball impact; T3: Follow-through phase: the phase was defined from impact until completion of the swing [33,34]. The specific phases are illustrated in Fig 2.

The racket's motion is recorded with a video camera and analyzed by Kinovea software to derive the vertical, and horizontal velocities of the racket head. The kinematic parameters provided racket tip velocity for the resultant horizontal and vertical velocity at impact, time duration, and average velocity in each phase.

All data were saved and analyzed using customized software (EMGworks 4.7.3 DelSys Inc.). In this study, we used the original EMG signals filtered by a 4th order Butterworth band-pass filter (50–500Hz) [35,36], and the amplitude analysis was carried out by root mean square (RMS) calculation. Before the experiment we tested the EMG data of the forehand full power hitting action of the athletes, the maximum activation values of each muscle were selected as maximal voluntary contractions (MVC) [32]. And all the data were standardized according to the MVC Processing. All EMG signals are taken from the moment of impact as the time zero point, with a time window of -500ms to + 1000ms. The EMG amplitude increases during the stroke, and the time window from -500ms to + 500ms is taken as the stroke phase, using the 'moment of touch ball' as the time zero point [37], Before the batting phase is the preparation phase and after the batting phase is the follow-through phase.

## Statistical analysis

Statistical analysis was performed using SPSS Statistics 24 software (IBM Corporation, Armonk, NY). Data normality was verified by using the Kolmogorov-Smirnov test. Independent t-tests were used to compare the kinematic differences between AG and IG in the three-movement phases at slow and fast serve speeds. The statistical significance level was set at 0.05 with $p$-values between 0.05 and 0.1 indicating a trend toward significance.

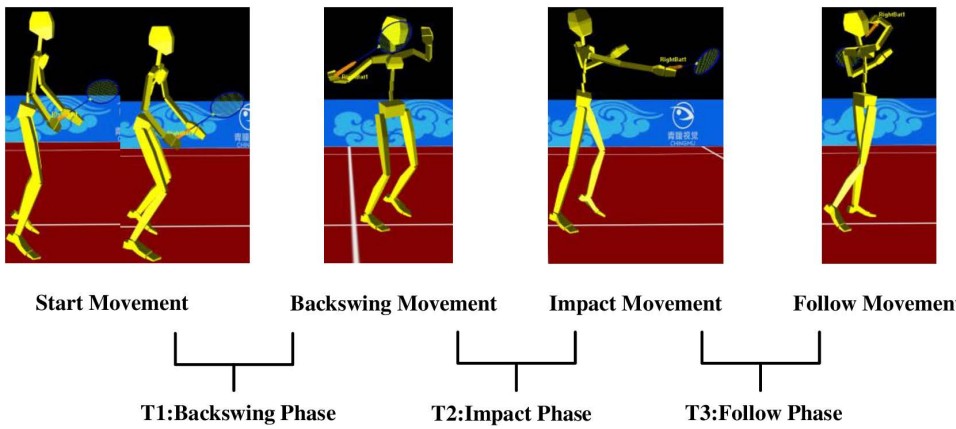

Start Movement    Backswing Movement    Impact Movement    Follow Movement

T1:Backswing Phase        T2:Impact Phase        T3:Follow Phase

**Fig 2. Three phases for the forehand receive motion.**

## Results

### The kinematics parameters of the racket tip

Table 2 shows the kinematic parameters of the racket head for AG and IG using the forehand receive technique at both serve speeds. The horizontal velocity, vertical velocity, and resultant velocity ($30.20 \pm 10.60$, $23.58 \pm 10.37$ and $21.52 \pm 9.27$m/s) at the moment of hitting the ball were significantly higher in AG than in the IG, at low serve speeds ($20.30 \pm 2.93$, $15.56 \pm 2.70$ and, $20.32 \pm 3.19$m/s). The resultant velocity ($18.29 \pm 4.94$m/s) was significantly higher than that of IG ($14.48 \pm 2.68$m/s) under high-speed serve, indicating that the swing speed of AG was faster than that of IG through-out the whole stroke phase. AG's racket speed decreased under the fast serve, comparing the IG, AG's return speed was faster and the return landing point was closer to the baseline. At the same time, as the serve speed increased, AG's return speed and placement scoring tended to increase, while IG's return speed and landing point tended to decrease, and increasing the swing speed affected the speed of the ball. However, AG showed a decrease in racket speed and an increase in return speed at high serve speeds, and there was no significant difference in the time taken for the three-movement phases of the forehand return technique between the two groups of participants, suggesting that the rate of muscle activation may have an effect on the effectiveness of stroke during the stroke.

### Muscle activation of the upper extremity at backswing

The results in Table 3 show the level of normalized activation of the right upper arm and trunk muscles during the backswing phase of the forehand receive in both groups of athletes. At low serve speeds, AG had significantly lower BC, DT, OEA, and LD normalized muscle activation during the backswing phase compared to IG. At high serve speeds, AG had significantly lower BC, DT, TP, and LD normalized muscle activation during the backswing phase compared to IG. In addition, the DT and TP of both groups of athletes had more involvement in the lead-in phase. The above results suggest that DT and TP on the right side have the main involvement in the lead-in phase, while BC, DT, and LD on the right side need to keep lower activation.

### Muscle activation of the upper extremity at import

The results in Table 4 show the degree of normalized activation of the right upper arm and trunk muscles in the impact phase using the forehand receive in both groups of athletes at both serve speeds. At low serve speeds, AG had significantly lower normalized activation of TB, BC, DT, TP, OEA, and LD during the stroke phase compared to IG. At high

**Table 2. Kinematic parameters of the racket tip.**

| Kinematics of the racket tip | Group | Lower-serve (Mean ±SD) | | | | | Higher-serve (Mean ±SD) | | | | |
|---|---|---|---|---|---|---|---|---|---|---|---|
| | | Variable | Bias-corrected 95%CI | | p | Effect size | Variable | Bias-corrected 95%CI | | p | Effect size |
| | | | Lower | Upper | | | | Lower | Upper | | |
| Horizontal velocity(m/s) | AG | 30.20±10.60 | 3.32 | 16.44 | 0.00* | 1.27 | 26.32±5.19 | -1.98 | 10.02 | 0.18 | 0.57 |
| | IG | 20.32±2.93 | 3.32 | 16.44 | | | 22.30±8.57 | -2.05 | 10.09 | | |
| Vertical velocity(m/s) | AG | 23.58±10.37 | 1.62 | 14.41 | 0.01* | 1.06 | 20.15±4.43 | -3.57 | 8.96 | 0.38 | 0.36 |
| | IG | 15.56±2.70 | 1.61 | 14.42 | | | 17.45±9.48 | -3.72 | 9.11 | | |
| Resultant velocity(m/s) | AG | 21.52±9.27 | 0.71 | 12.39 | 0.03* | 0.94 | 18.29±4.94 | 0.44 | 7.17 | 0.02* | 0.96 |
| | IG | 14.97±3.19 | 0.72 | 12.37 | | | 14.48±2.68 | 0.38 | 7.22 | | |
| Backswing(s) | AG | 0.94±0.22 | -0.01 | 0.29 | 0.07 | 0.62 | 0.68±0.18 | -0.14 | 0.15 | 0.91 | 0.05 |
| | IG | 0.80±0.23 | -0.01 | 0.28 | | | 0.67±0.21 | -0.15 | 0.16 | | |
| Impact(s) | AG | 0.45±0.15 | -0.09 | 0.12 | 0.75 | 0.13 | 0.40±0.07 | -0.06 | 0.08 | 0.75 | 0.11 |
| | IG | 0.43±0.16 | -0.09 | 0.12 | | | 0.39±0.11 | -0.07 | 0.09 | | |
| Follow-through(s) | AG | 0.99±0.19 | -0.06 | 0.20 | 0.28 | 0.34 | 0.48±0.14 | -0.09 | 0.09 | 0.96 | 0.00 |
| | IG | 0.92±0.22 | -0.06 | 0.20 | | | 0.48±0.10 | -0.09 | 0.08 | | |
| Return velocity (km/h) | AG | 132.98±31.97 | 4.93 | 42.95 | 0.01* | 0.90 | 136.33±26.16 | 11.86 | 48.04 | 0.00** | 1.26 |
| | IG | 109.04±19.97 | 2.15 | 45.73 | | | 106.38±21.26 | 10.91 | 48.99 | | |
| Ball placement score | AG | 2.83±0.58 | 0.33 | 1.02 | 0.03* | 1.38 | 2.92±0.00 | 0.83 | 1.41 | 0.00** | 3.00 |
| | IG | 2.16±0.37 | 0.28 | 1.07 | | | 1.88±0.49 | 0.87 | 1.37 | | |

Note: *p<0.05; AG, advanced group; IG, intermediate group.

**Table 3. Normalized activation of right upper arm and trunk muscles in forehand receives at backswing (% MVC).**

| Muscle | Group | Lower-serve receive | | | | | Higher-serve receiv Bias-corrected 95%CI e | | | | |
|---|---|---|---|---|---|---|---|---|---|---|---|
| | | Variable | | | p | Effect size | Variable | Bias-corrected 95%CI | | p | Effect size |
| | | | Lower | Upper | | | | Lower | Upper | | |
| BB | AG | 2.94±3.66 | -2.60 | 1.98 | 0.78 | 0.08 | 2.79±2.67 | -4.37 | 1.08 | 0.23 | 0.32 |
| | IG | 3.26±4.15 | -2.63 | 2.00 | | | 4.43±6.71 | -4.62 | 1.34 | | |
| TB | AG | 3.81±2.50 | -4.49 | 0.21 | 0.07 | 0.53 | 6.14±6.38 | -5.73 | 3.15 | 0.56 | 0.16 |
| | IG | 5.95±5.19 | -4.62 | 0.34 | | | 7.43±9.65 | -5.94 | 3.36 | | |
| BC | AG | 4.39±3.33 | -7.66 | -2.38 | 0.00** | 1.10 | 5.67±3.16 | -5.75 | -1.58 | 0.00** | 0.96 |
| | IG | 9.41±5.52 | -7.77 | -2.27 | | | 9.34±4.38 | -5.83 | -1.51 | | |
| DT | AG | 10.38±7.82 | -14.63 | -1.95 | 0.01* | 0.76 | 8.48±4.60 | -12.84 | -3.02 | 0.00** | 0.86 |
| | IG | 18.68±13.36 | -14.90 | -1.68 | | | 16.42±12.18 | -13.31 | -2.55 | | |
| TP | AG | 13.14±6.58 | -11.77 | 1.49 | 0.12 | 0.45 | 10.31±5.43 | -14.17 | -2.85 | 0.00** | 0.80 |
| | IG | 18.28±14.91 | -12.18 | 1.91 | | | 18.82±13.98 | -14.70 | -2.32 | | |
| PM | AG | 2.23±2.74 | -4.06 | 0.44 | 0.11 | 0.46 | 2.70±3.15 | -3.47 | 0.65 | 0.18 | 0.37 |
| | IG | 4.04±4.78 | -4.16 | 0.54 | | | 4.10±4.31 | -3.54 | 0.72 | | |
| OEA | AG | 5.62±3.75 | -4.78 | -0.43 | 0.02* | 0.70 | 9.11±8.55 | -9.08 | 1.28 | 0.14 | 0.35 |
| | IG | 8.22±3.63 | -4.77 | -0.43 | | | 13.01±10.24 | -9.19 | 1.38 | | |
| LD | AG | 5.60±3.00 | -17.40 | -5.85 | 0.00** | 1.15 | 9.5±7.70 | -20.71 | -7.06 | 0.00** | 1.09 |
| | IG | 17.22±14.00 | -17.93 | -5.32 | | | 23.38±16.25 | -21.25 | -6.52 | | |

Note: *p<0.05; AG, advanced group; IG, intermediate group.

**Table 4. Normalized activation of right upper arm and trunk muscles in forehand receives at impact (% MVC).**

| Muscle | Group | Lower-serve receive | | | | | Higher-serve receive | | | | |
|---|---|---|---|---|---|---|---|---|---|---|---|
| | | Variable | Bias-corrected 95%CI | | p | Effect size | Variable | Bias-corrected 95%CI | | p | Effect size |
| | | | Lower | Upper | | | | Lower | Upper | | |
| BB | AG | 22.43±14.91 | -20.55 | 4.75 | 0.21 | 0.36 | 24.06±13.67 | -21.37 | -0.83 | 0.04* | 0.58 |
| | IG | 30.33±27.11 | -21.16 | 5.35 | | | 35.17±23.12 | -21.97 | -0.23 | | |
| TB | AG | 17.98±9.19 | -38.67 | -14.12 | 0.00** | 1.23 | 23.08±15.06 | -33.20 | -8.85 | 0.00** | 0.93 |
| | IG | 44.37±28.89 | -39.64 | -13.15 | | | 44.11±28.19 | -34.03 | -8.02 | | |
| BC | AG | 28.96±10.16 | -29.04 | -6.45 | 0.00** | 0.90 | 28.07±9.84 | -32.57 | -12.57 | 0.00** | 1.21 |
| | IG | 46.7±25.89 | -29.82 | -5.67 | | | 50.64±24.59 | -33.49 | -11.64 | | |
| DT | AG | 24.25±11.86 | -26.88 | -6.52 | 0.00** | 0.95 | 23.01±14.18 | -25.29 | -6.98 | 0.00** | 0.96 |
| | IG | 40.95±21.9 | -27.38 | -6.03 | | | 39.15±19.00 | -25.60 | -6.67 | | |
| TP | AG | 32.16±18.37 | -29.46 | -0.84 | 0.04* | 0.61 | 32.27±16.65 | -23.45 | -1.68 | 0.03* | 0.63 |
| | IG | 47.31±29.67 | -30.02 | -0.29 | | | 44.83±22.78 | -23.84 | -1.29 | | |
| PM | AG | 32.3±13.69 | -22.87 | 1.44 | 0.08 | 0.51 | 31.22±15.56 | -34.55 | -11.72 | 0.00** | 1.09 |
| | IG | 43.01±26.44 | -23.51 | 2.07 | | | 54.36±25.43 | -35.18 | -11.09 | | |
| OEA | AG | 24.37±8.07 | -29.00 | -11.72 | 0.00** | 1.35 | 25.21±9.25 | -29.54 | -12.75 | 0.00** | 1.35 |
| | IG | 44.73±19.68 | -29.58 | -11.14 | | | 46.36±20.13 | -30.23 | -12.06 | | |
| LD | AG | 19.55±9.63 | -23.90 | -7.02 | 0.00** | 1.06 | 23.84±15.98 | -30.01 | -8.06 | 0.00** | 0.94 |
| | IG | 35.01±18.29 | -24.33 | -6.59 | | | 42.87±23.65 | -30.49 | -7.58 | | |

Note: *p < 0.05; AG, advanced group; IG, intermediate group.

serve speeds, the normalized activation of all eight muscles was lower in the stroke phase in AG than in IG, and TP and PM had more involvement in the stroke phase in AG at both serve speeds, whereas BC and TP had more involvement in IG at low serve speeds, and BC and PM had more involvement in IG at high serve speeds. The above results showed that AG's TP and PM are more involved during the stroke phase, and other muscles kept lower involvement, while the main force-generating muscle groups in IG were not constant, and the activation level of each muscle group tended to be the same.

### Muscle activation of the upper extremity at follow-through

The results in Table 5 show the normalized activation levels of the right upper arm and trunk muscles in the follow-through using forehand receive at two serve speeds for both groups of athletes. At low serve speeds, the activation of BC and OEA in the follow-through was significantly lower in AG than in IG. At high serve speeds, the activation of BC in the follow-through was significantly lower in AG than in IG. In addition, the DT and TP in both groups of athletes showed more engagement in the follow-through. The normalized activation of the right upper arm and trunk muscles was significantly lower in AG than in IG.

## Discussion

The purpose of this study was to investigate the differences in activation strategies and stroke performance of the right upper arm and trunk muscles of high-level and intermediate athletes using a forehand receive maneuver at two serve speeds. It was found that high-level athletes had better stroke performance (faster return speeds and higher return scores) and that their return speeds and placement scores tended to increase with faster serve speeds, whereas intermediate athletes showed a decreasing trend. In terms of muscle activation, high-level athletes showed low activation rates

**Table 5. Normalized activation of right upper arm and trunk muscles in forehand receives at follow-through (% MVC).**

| Muscle | Group | Lower-serve receive | | | | | Higher-serve receive | | | | |
|---|---|---|---|---|---|---|---|---|---|---|---|
| | | Variable | Bias-corrected 95%CI | | p | Effect size | Variable | Bias-corrected 95%CI | | p | Effect size |
| | | | Lower | Upper | | | | Lower | Upper | | |
| BB | AG | 2.43±2.97 | -5.22 | 1.71 | 0.31 | 0.48 | 2.79±3.57 | -5.33 | 0.90 | 0.16 | 0.38 |
| | IG | 4.18±8.01 | -5.47 | 1.96 | | | 5.00±7.40 | -5.57 | 1.15 | | |
| TB | AG | 5.12±6.73 | -1.06 | 4.97 | 0.20 | 0.39 | 5.65±6.09 | -8.66 | 1.81 | 0.19 | 0.35 |
| | IG | 3.16±2.14 | -0.95 | 4.86 | | | 9.08±12.37 | -9.06 | 2.21 | | |
| BC | AG | 2.78±2.36 | -5.91 | -0.43 | 0.02* | 0.70 | 2.77±2.64 | -4.13 | -0.70 | 0.00** | 0.77 |
| | IG | 5.95±6.33 | -6.11 | -0.23 | | | 5.19±3.57 | -4.19 | -0.64 | | |
| DT | AG | 8.09±6.08 | -9.85 | 1.00 | 0.11 | 0.47 | 7.77±8.16 | -9.57 | 1.71 | 0.17 | 0.38 |
| | IG | 12.51±11.82 | -10.13 | 1.29 | | | 11.69±12.20 | -9.82 | 1.97 | | |
| TP | AG | 10.5±8.68 | -10.08 | 3.26 | 0.31 | 0.30 | 10.22±7.16 | -7.80 | 3.64 | 0.50 | 0.20 |
| | IG | 13.9±13.75 | -10.33 | 3.51 | | | 12.3±13.19 | -8.18 | 4.02 | | |
| PM | AG | 6.12±5.89 | -2.39 | 3.92 | 0.62 | 0.14 | 6.62±6.75 | -6.09 | 1.96 | 0.31 | 0.28 |
| | IG | 5.35±4.68 | -2.34 | 3.87 | | | 8.69±7.84 | -6.16 | 2.03 | | |
| OEA | AG | 5.25±4.36 | -5.86 | -0.53 | 0.02* | 0.70 | 7.49±7.05 | -5.31 | 1.75 | 0.32 | 0.28 |
| | IG | 8.45±4.72 | -5.88 | -0.51 | | | 9.28±5.42 | -5.22 | 1.66 | | |
| LD | AG | 6.69±5.06 | -1.72 | 3.56 | 0.50 | 0.20 | 6.91±4.99 | -8.84 | 2.48 | 0.26 | 0.30 |
| | IG | 5.78±3.72 | -1.67 | 3.51 | | | 10.09±14.18 | -9.41 | 3.05 | | |

Note: *p<0.05; AG, advanced group; IG, intermediate group.

for all muscle groups and clear dominant muscles in all phases compared to intermediate-level athletes. Differences in muscle utilization between the two groups of athletes and the effect of the above differences on stroke performance are emphasized.

During the Backswing phase, the muscle activity characteristics of AG and IG at both serve speeds were more similar and in agreement with the results of the Abuwarda study [17]. It was observed that DT (10.38, 18.68%) and TP (13.14, 18.28%) of both groups of athletes showed higher muscle activity since the deltoid muscle produces the shoulder toward the spine by stretching and horizontal extension. Force of backward flexion, TP is responsible for bringing the scapula toward the spine, and the upper arm performs a backward flexion movement, indicating that, the leading movement is mainly accomplished by turning the shoulder and backward extension of the upper arm during the leading phase [38]. Comparing the differences in muscle activity between the two groups of athletes, the overall muscle activity of the AG was low compared to the IG, with significant differences in BC, DT, and LD activity between the two groups. BC was responsible for the flexion and extension of the elbow joint, producing the force of forearm adduction and abduction. More consistent with the DT and TP roles, the LD was responsible for flexion and extension of the shoulder joint to accomplish the backswing of the upper arm. Based on the above results, it can be seen that in the Backswing phase, both groups of athletes, led through shoulder rotation and rapid backward extension of the upper arm to complete the backswing phase, while the IG showed more forearm abduction movements, which may increase the length of the backswing phase, affecting the preparation of the next phase of the movement.

During the Impact phase, there were differences in muscle activity characteristics between the two groups of athletes, with AG's muscle activities being more similar at both serve speeds, whereas IG's muscle activities were unstable at both serve speeds. The activity of each muscle was significantly lower in AG compared to IG. The highest TP (32.16, 32.27%) and PM (32.30, 31.22%) activity was observed for AG at both serve speeds and there was no difference in the amount of activity, with both TP and PM generating the force of shoulder forward flexion [38]. It shows that AG accomplishes forward

stroke with TP and PM of the trunk as the dominant muscles. The muscle activity characteristics of IG are different from AG. It is observed that IG has the highest activity of BC (46.7%) and TP (47.31%) at low serve speed. When the incoming ball speed becomes faster, IG has the highest activity of BC (50.64%) and PM (54.36%). Both TP and PM generate forces of forward shoulder flexion, while BC is responsible for elbow flexion and extension and generates forces of forearm pronation, suggesting that the IG has more forearm action than the AG when striking the ball forward. More forearm action is likely to maintain wrist stability, which retains the stroke characteristics of a forehand groundstroke [14]. The tennis forehand stroke is a bottom-up whipping action, with trunk force and acceleration of the upper limbs in all segments, which acts on the racket through the wrist joints [39]. This is consistent with our findings that the AG has a faster racket head speed. The IG increases the amount of forearm activity in the process, which interrupts the kinetic energy transmitted by the trunk and acts on the racket with reduced speed so that the IG return speed is lower as the speed of the serve increases. Differences in the activation of trunk flexors and extensors can appear as a tennis-specific power imbalance [40]. Roetert believes that High-level tennis players have above-average trunk extensor strength [41]. Consistently, AG has higher activation of the OEA at both serve speeds, suggesting that high-level tennis players are more adept at generating power through the trunk. In contrast, the faster the AG returns the ball with trunk power to achieve acceleration of the upper limbs. These results provide more information about muscle activation around the shoulder during the stroke phase, which may benefit researchers, coaches, and tennis players in preventing shoulder injuries and improving stroke performance.

During the Follow-through phase, the muscle activity levels of the two groups of athletes were characterized more similarly. It can be observed that DT (8.09, 12.51%) and TP (10.50, 13.9%) had greater activity in AG and IG at low high-speed serves, and DT (7.77, 11.69%) and TP (10.22, 12.30%) had greater activity in AG and IG at high-speed serves. The follow-through motion is generated by the inertia after a quick stroke, and the involvement of DT and TP is to stabilize the motion during the follow-through. Compared to the IG, the BC activity of the AG was significantly lower in the follow-through phase of the swing at both serve speeds, with the BC being responsible for elbow flexion and extension [38], and the BC showed lower activity, suggesting that there was less pronation of the forearm in the AG.

There are some limitations in this study. First, the spin speed of the return was not tested in the test, although the focus of the study was on the EMG changes of the athletes' receive movements under different serving conditions, the combination of the spin speed can be more objectively evaluated on the return; second, only male athletes were selected as the test subjects in the test, and due to the gender difference there will be the difference of strength and other physical conditions, the significant difference between the male and female athletes' receive performances under the same serving conditions may be more than technical differences, which may cause ambiguous conclusions, and the receive performance of female athletes will be further investigated in future studies. Thirdly, high-speed serves lead to faster muscle fatigue, which in turn affects activation patterns or athletic performance. Due to the 3-minute interval between each set of serve tests, the fatigue effect was not tested in the present study, and subsequent studies should consider the effect of fatigue mechanisms on serves. Fourth, only the athlete's right trunk and arm muscles were selected for this study, muscle activation of the left upper limb and part of the trunk were excluded from the present analysis.

## Conclusion

This study investigated the differences in activation of right trunk and upper limb muscle groups during forehand receive at two serve speeds in athletes of different levels. We found that, compared to IG, AG has a clear dominant muscle group at each stage, along with higher muscle group activation and significantly lower activation of other muscle groups. TP plays an important role in the backswing phase. TP and PM are the leading force-generating muscles in the stroke phase, while the activation of other muscles should be reduced. During the follow-through phase of the swing, DT and TP are the leading muscles in keeping the follow-through movement stable. Therefore, when the serve speed becomes faster, players keep their muscle activation lower before the stroke, facilitating smoother force transmission while preventing injuries. In

the future, lower limb muscle groups, left side muscle groups, female participants, and ball rotation should be considered to fully explore the biomechanical characteristics of tennis forehand receive the technique.

## Supporting information

**S1 File. Original data.**
(XLSX)

## Acknowledgments

We want to thank all the participants and their coaches for their support in this study.

## Author contributions

**Conceptualization:** Yuxin Zhang, Zhouye Chen.

**Data curation:** Yuxin Zhang, Jiajie Tian.

**Investigation:** Yuxin Zhang, Jiajie Tian.

**Project administration:** Yuxin Zhang.

**Validation:** Zhouye Chen.

**Writing – original draft:** Yuxin Zhang, Jiajie Tian.

**Writing – review & editing:** Yuxin Zhang, Zhouye Chen.

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
