## [Decision Letter · Decision Letter 0]

10 Dec 2024

PONE-D-24-50212The differences of muscle activation in forehand serve-receiving technique of male tennis players at different skillsPLOS ONE

Dear Dr. Chen,

Thank you for submitting your manuscript to PLOS ONE. After careful consideration, we feel that it has merit but does not fully meet PLOS ONE’s publication criteria as it currently stands. Therefore, we invite you to submit a revised version of the manuscript that addresses the points raised during the review process.

We look forward to receiving your revised manuscript.

Kind regards,

Yaodong Gu

Academic Editor

PLOS ONE

Journal Requirements:

Reviewers' comments:

Reviewer's Responses to Questions

**Comments to the Author**

1. Is the manuscript technically sound, and do the data support the conclusions?

Reviewer #1: Yes

Reviewer #2: Partly

2. Has the statistical analysis been performed appropriately and rigorously? 

Reviewer #1: Yes

Reviewer #2: Yes

3. Have the authors made all data underlying the findings in their manuscript fully available?

Reviewer #1: Yes

Reviewer #2: Yes

4. Is the manuscript presented in an intelligible fashion and written in standard English?

Reviewer #1: Yes

Reviewer #2: Yes

5. Review Comments to the Author

Reviewer #1: Reviewer comments

This manuscript entitled “The differences of muscle activation in forehand serve-receiving technique of male tennis players at different skills” the relationship between muscle activation patterns and athletic performance in high-level (AG) and intermediate-level (IG) athletes at different serve speeds when performing forehand return movements.

Specific comments are shown below:

Abstract:

1. Lines 15-17: The abbreviations (AG, IG) are not explicitly stated in the body of the article and are recommended to be stated when they first appear. For example, "30 male tennis players (no difference in age, height, and weight) were divided into senior (AG, n=15) and intermediate (IG, n=15) groups......"

2. Lines 24-25: The "three phases" do not specify which phases (e.g. before, during, and after the shot) and it is recommended to add a clear definition.

Introduction:

1. Line 32: The current literature review on muscle activation research is relatively general, citing a wide range of studies (e.g., grip, path, rotation, etc.), and the authors can focus on the blank spots in the study of muscle activation in the return ball.

2. Lines 62-63: Errors in the text should be corrected promptly to ensure that all references are linked accurately.

Methods:

1. Lines 91-92: Specific information for statistical power analyses (e.g., effect size) can be added to demonstrate the adequacy of the sample size.

2. Lines 127-129: Although the options for wide-court zones and T-shaped/wide-court serves are mentioned, the rationale is brief. The author can provide more background information on how these settings simulate real-world match scenarios.

Lines 159-163: The EMG signal data processing needs to provide more details on the muscle activation solution is taken as a second-order differential equation? To provide more effective evidence, the authors may consider referring to the following relevant studies: Accurately and effectively predict the ACL force: Utilizing biomechanical landing pattern before and after-fatigue (https://doi.org/10.1016/j.cmpb.2023.107761); Adaptive neuro-fuzzy inference system model driven by the non-negative matrix factorization-extracted muscle synergy patterns to estimate lower limb joint movements (https://doi.org/10.1016/j.cmpb.2023.107848)

Results:

1. Lines 218-221: "TP and PM are the main muscle groups, and these are part of the discussion. The results section should be dominated by data presentations and avoid speculative or explanatory language.

Discussion:

1. Line 252: Check and fix reference errors.

2. Lines 310-311: The high-speed serve was studied to lead to faster muscle fatigue, which in turn affected activation patterns or athletic performance. If fatigue effects are not tested, their possible effects should be mentioned in the limitations.

Conclusion:

1. In the conclusion section, the practical application of the research and the future research direction can be explained more clearly.

Reviewer #2: Dear Authors,

the Authors did a good job in writing a readable manuscript. This paper could be considered qualified to be published on “PlosOne” if the authors apply the modifications requested, "Major issues" are recommended. It is the reason why "Major revision" is my personal choice. I would be happy to review a revised version of this manuscript.

The aim of the submitted manuscript was to analyze differences in muscle activation of the right upper limb and part of the trunk and differences in stroke performance of forehand serve receiving in male tennis players of different levels of performance at different speed.

This paper could be considered qualified to be published if the authors apply the modifications requested.

While it is a very interesting topic, some suggestions for improving the paper are provided below:

• The basic descriptive level adopted by author(s) does not seem compatible with interests of both researchers' biomechanical conceptual challenges and coaches' practical applications to training. The main concern refers to the weak/absent theoretical background; there is no perceivable research question or rationale for the study presented to readers.

Author(s) do not consider any specific difficulties or other aspects of performing these different types of tennis techniques to minimally justify the comparison; on the other hand, the use of the descriptive knowledge obtained from the observed differences was not discussed in terms of applications for athletes and coaches.

In short, the rationale for the study is unclear. The manuscript is limited to a biomechanical laboratory exercise to spatially and temporally describe the participants' movement.

• It is important to include more information about the balls. It is not possible to totally exclude these information from the paper (brand, company, model, etc).

• I think much of the Procedure section needs to be rewritten to better describe procedures. I would suggest dividing into subchapters (Examples: EMG, warm-up, set-up, etc.)

• The following variables can’t be totally excluded from the paper: number of repetitions, serve errors, serves in, let, etc.

• Figure 1 shows the Serve-return Motion Capture setup diagram. Dimension/position of Serve placement area and dimension/position of the Serve placement area have to be justified by Authors. References about this setup need to be included to support this methodology.

• Line 124. There is a typing error (Table tennis?)

• Limitations: muscle activation of the left upper limb and part of the trunk were excluded from the present analysis

Finally, this manuscript could be considered qualified to be published if the authors apply the previous modifications, therefore “Major revision” is recommended.

6. PLOS authors have the option to publish the peer review history of their article (what does this mean? ). If published, this will include your full peer review and any attached files.

**Do you want your identity to be public for this peer review?** For information about this choice, including consent withdrawal, please see our Privacy Policy .

Reviewer #1: No

Reviewer #2: No

---

## [Decision Letter · Decision Letter 1]

23 Mar 2025

PONE-D-24-50212R1The differences of muscle activation in forehand serve-receiving technique of male tennis players at different skillsPLOS ONE

Dear Dr. Chen,

Thank you for submitting your manuscript to PLOS ONE. After careful consideration, we feel that it has merit but does not fully meet PLOS ONE’s publication criteria as it currently stands. Therefore, we invite you to submit a revised version of the manuscript that addresses the points raised during the review process.

We look forward to receiving your revised manuscript.

Kind regards,

Yaodong Gu

Academic Editor

PLOS ONE

Reviewers' comments:

Reviewer's Responses to Questions

**Comments to the Author**

1. If the authors have adequately addressed your comments raised in a previous round of review and you feel that this manuscript is now acceptable for publication, you may indicate that here to bypass the “Comments to the Author” section, enter your conflict of interest statement in the “Confidential to Editor” section, and submit your "Accept" recommendation.

Reviewer #1: (No Response)

Reviewer #3: (No Response)

2. Is the manuscript technically sound, and do the data support the conclusions?

Reviewer #1: Yes

Reviewer #3: Partly

3. Has the statistical analysis been performed appropriately and rigorously? 

Reviewer #1: Yes

Reviewer #3: N/A

4. Have the authors made all data underlying the findings in their manuscript fully available?

Reviewer #1: Yes

Reviewer #3: No

5. Is the manuscript presented in an intelligible fashion and written in standard English?

Reviewer #1: Yes

Reviewer #3: Yes

6. Review Comments to the Author

Reviewer #1: Thank you for your revised manuscript. Upon review, I noticed that the suggested references and additional review points have not been fully addressed. To ensure a more comprehensive discussion and strengthen the manuscript’s contribution, I encourage you to carefully incorporate the recommended references and revisions. Addressing these points will improve the clarity and rigor of your study.

Please let me know if any clarification is needed. Looking forward to your updated submission.

Reviewer #3: The study has significant flaws in experimental design, data collection, and analysis, compromising its validity and reliability. Using a manual serve instead of standardized equipment introduces subjectivity and reduces reproducibility. The reliance on a single high-speed camera limits accurate three-dimensional kinematic analysis, with no calibration or error analysis reported. EMG data processing lacks transparency, with incomplete details on noise filtering, baseline correction, and temporal characterization, resulting in a limited understanding of muscle synergy. Additionally, the absence of effect sizes and confidence intervals weakens the statistical analysis. To enhance scientific rigor, the authors should adopt standardized equipment, improve data collection methods, and ensure proper statistical reporting.

7. PLOS authors have the option to publish the peer review history of their article (what does this mean? ). If published, this will include your full peer review and any attached files.

**Do you want your identity to be public for this peer review?** For information about this choice, including consent withdrawal, please see our Privacy Policy .

Reviewer #1: No

Reviewer #3: No

---

## [Decision Letter · Decision Letter 2]

14 Apr 2025

The differences of muscle activation in forehand serve-receiving technique of male tennis players at different skills

PONE-D-24-50212R2

Dear Dr. Chen,

We’re pleased to inform you that your manuscript has been judged scientifically suitable for publication and will be formally accepted for publication once it meets all outstanding technical requirements.

Kind regards,

Yaodong Gu

Academic Editor

PLOS ONE

Additional Editor Comments (optional):

Reviewers' comments:

Reviewer's Responses to Questions

**Comments to the Author**

1. If the authors have adequately addressed your comments raised in a previous round of review and you feel that this manuscript is now acceptable for publication, you may indicate that here to bypass the “Comments to the Author” section, enter your conflict of interest statement in the “Confidential to Editor” section, and submit your "Accept" recommendation.

Reviewer #1: All comments have been addressed

Reviewer #3: All comments have been addressed

2. Is the manuscript technically sound, and do the data support the conclusions?

Reviewer #1: Yes

Reviewer #3: Yes

3. Has the statistical analysis been performed appropriately and rigorously? 

Reviewer #1: Yes

Reviewer #3: Yes

4. Have the authors made all data underlying the findings in their manuscript fully available?

Reviewer #1: Yes

Reviewer #3: Yes

5. Is the manuscript presented in an intelligible fashion and written in standard English?

Reviewer #1: Yes

Reviewer #3: Yes

6. Review Comments to the Author

Reviewer #1: (No Response)

Reviewer #3: The authors have carefully responded to the review comments one by one, and have made corresponding revisions and additions to the original manuscript, including the explanation of the setting of serving conditions, the clarification of the purpose of using high-speed cameras, the improvement of the EMG data processing method, and the addition of effect sizes and confidence intervals in the statistical results. After comprehensive evaluation, the manuscript has been significantly improved in terms of scientificity and standardization.

7. PLOS authors have the option to publish the peer review history of their article (what does this mean? ). If published, this will include your full peer review and any attached files.

**Do you want your identity to be public for this peer review?** For information about this choice, including consent withdrawal, please see our Privacy Policy .

Reviewer #1: No

Reviewer #3: No

---

## [Editor Report · Acceptance letter]

PONE-D-24-50212R2

PLOS ONE

Dear Dr. Chen,

I'm pleased to inform you that your manuscript has been deemed suitable for publication in PLOS ONE. Congratulations! Your manuscript is now being handed over to our production team.

Kind regards,

on behalf of

Professor Yaodong Gu

Academic Editor

PLOS ONE